# Temporal-Spatial Evolution and Trend Prediction of the Supply Efficiency of Primary Medical Health Service—An Empirical Study Based on Central and Western Regions of China

**DOI:** 10.3390/ijerph20031664

**Published:** 2023-01-17

**Authors:** Fang Wu, Mingyao Gu, Chenming Zhu, Yingna Qu

**Affiliations:** School of International Pharmaceutical Business, China Pharmaceutical University, Nanjing 211198, China

**Keywords:** primary medical health service, spatial-temporal evolution, super efficiency Slack-Based Measure model, Markov chain analysis, efficiency

## Abstract

China has established a comprehensive primary medical health service system, but the development of primary medical health services in the central and western regions is still unbalanced and insufficient. Based on data from 2010 to 2019, this paper constructs a super efficiency Slack-Based Measure model to calculate the supply efficiency of primary medical health services in 20 provinces and cities in central and western China. Using Kernel density estimation and Markov chain analysis, this paper further analyzes the spatial-temporal evolution of the supply efficiency of primary medical health services in central and western China, and also predicts the future development distribution through the limiting distribution of Markov chain to provide a theoretical basis for promoting the sinking of high-quality medical resources to the primary level. The results show that firstly, during the observation period, the center of the Kernel density curve moves to the left, and the main peak value decreases continuously. The main diagonal elements of the traditional Markov transition probability matrix are 0.7872, 0.5172, 0.8353, and 0.7368 respectively, which are significantly larger than other elements. Secondly, when adjacent to low state and high state, it will develop into convergence distributions of 0.7251 and 0.8243. The supply efficiency of primary medical health services in central and western China has the characteristics of high (Ningxia) and low (Shaanxi) aggregation respectively, but the aggregation trend is weakened. Thirdly, the supply efficiency of health services has the stability of keeping its own state unchanged, but the transition of state can still occur. The long-term development of the current trend cannot break the distribution characteristics of the high and low clusters, the efficiency will show a downward trend in the next 10–20 years, and still the problem of uneven long-term development emerges.

## 1. Introduction

In 1978, organized by the World Health Organization and the United Nations International Children’s Emergency Fund, the World Health Assembly issued the Declaration of Almaty, introducing the concept of “primary healthcare” for the first time, which is considered to be the strategic goal and basic approach to achieve universal access to health services by the year 2000 [1]. In the decades following, the global medical technology has been greatly improved, and primary health care systems have shown greater effectiveness, higher efficiency, and much more equity. However, on a global scale, there is a huge gap in the development of primary healthcare between developed and developing countries, and most countries also have unbalanced development of primary healthcare due to the unbalanced economic development in different areas [2]. As a developing country, this problem has also occurred in China, so the Chinese government has launched the new medical reform in 2009 to build and strengthen the primary medical health services system [3].

In early 2020, COVID-19 was raging. Primary medical health institutions played the role of the bottom of the defending network as sentinels for epidemic prevention and control [4]. The efficient operation of primary medical and health institutions contributed to detect, isolate, report, and refer patients or suspected patients in time, making them an important force in epidemic prevention and control [5]. On 2 July 2020, the Chinese government issued “The Guiding Opinions of the CPC Central Committee and The State Council on Promoting the Development of Western China and Forming a New Pattern in the New Era”, proposing to improve the capacity and level of medical services in western China, accelerate the standardization of primary medical health care, and support Ningxia in building an “Internet plus Medical and health” demonstration zone. In December of the following year, the Chinese government issued “the CPC Central Committee and the State Council on Promoting High-quality Development in the Central Region in the New Era”, emphasizing that the central region should give full play to its regional advantages of “Connecting the East and the West” and “Connecting the South and the North”, so as to promote the development of high-quality service of primary medical and health institutions. It can be seen that the Chinese government attaches great importance to the development and construction of primary medical health care in the central and western regions.

In order to control drug costs and emphasize the public benefit attribute of hospitals, the Chinese government pledged in 2017 to eliminate drug addiction, which means that public hospitals at the county level and above would eliminate drug addiction and sell some drugs at near-cost prices. According to the data of China’s National Bureau of Statistics, after the abolition of drug addiction, the utilization rate of hospital beds in primary medical health institutions decreased from 58.1% to 53.6% in 2020 compared with 2017, the average length of stay increased from 7.9 days to 8.1 days, and the proportion of residents’ medical expenses decreased from 40.9% to 36.9% [6]. The decline in the proportion of medical expenses indicated that the efficiency of primary medical health services had improved, but the decline in the utilization rate of hospital beds and the increased length of stay in hospital indicated that the primary medical health resources were not fully utilized, and the medical resources invested in by the government had not been effectively utilized [7]. From the perspective of the eastern, central, and western regions, the distribution of medical resources in China is still unbalanced. In 2020, the number of primary health personnel in the eastern, central, and western regions accounted for about 42.47%, 29.15%, and 28.77%, respectively, and the number of treatment visits accounted for about 47.91%, 27.81%, and 24.28%. The development level of the central and western regions were lower than that of the eastern regions, and the differences in natural and social environments was significant, which led to the disadvantage of technology and resources in the allocation of primary medical resources in the central and western regions [8]. The supply of medical resources in the central and western regions are relatively scarce, the utilization efficiency is not high, and the technical levels still need to be improved, which makes the supply of primary medical resources more difficult. Coupled with the impact of the epidemic, the problem of low service efficiency is particularly severe [9]. In addition, the current research focuses on the developed eastern regions and public hospitals, and scholars’ research on the efficiency of primary healthcare in central and western China is insufficient. Therefore, how is the supply level of primary medical health services in central and western China? What are the spatial-temporal distribution characteristics of the research results? How is the future development trend of primary medical health service supply in central and western China? These issues need to be further studied and discussed.

Previous articles on the supply efficiency of health service have mostly focused on the economic impact of human resource efficiency [10] and the impact of human resource efficiency and medical institutions [11]. Researchers have mostly measured the supply efficiency of Chinese health services from a national perspective [12] or evaluated the systemic level of health care in China [13]. Therefore, in the context of Chinese policies, existing gaps, and cutting-edge research, what is the level of supply efficiency of primary health services in central and western China? What are the temporal and spatial distribution characteristics of the study results? What are the trends in the future development of supply efficiency of primary health services in central and western China? These questions require further research and discussions. According to this, based on the panel data of National Bureau of Statistics of China, this paper applies super efficiency Slack-Based Measure to analyze the difference in supply efficiency of health services, and further combines it with Kernel density estimation and Markov chain analysis to reveal the development nature of supply efficiency of primary health services in 20 provinces of central and western China. The aim is to provide effective reference for promoting the sinking of high-quality medical resources to the primary level and the construction of primary medical health system.

## 2. Literature Review

There is no unified and authoritative definition of the supply efficiency of medical health services. The definition of supply efficiency has first been put forward by Samuelson when studying public goods; that is, the supply efficiency of public goods means that the supply of public goods reaches Pareto Optimal [14]. Charnes [15] has understood supply efficiency as the optimal combination of input factors when output was given. Sengupta [16] defined the technical efficiency of health services as the optimal combination of factors of production and the management to meet consumer demand of health services under given financial conditions. WHO [17] defines it as the attempt to obtain the maximum health output for the health resources invested. Combined with existing studies, this paper defines the supply efficiency of primary medical health services as the maximum of health services provided by primary medical health institutions with limited health resources.

Quantitative research on primary medical health care mainly focuses on two aspects: efficiency measurement and spatial analysis. At present, the methods to measure the efficiency of primary healthcare include the parametric method and the non-parametric method. The parameter method represented by the stochastic frontier model is used to revise the weight definition and measure the efficiency [18]. However, researchers’ subjective factors will bias the results when the weight is defined, resulting in the deviation of the results [19]. Therefore, more and more studies have adopted non-parametric estimation based on Data Envelopment Analysis to measure the efficiency of primary healthcare. DEA can be used to study the input-output of multiple decision-making units at the same time without weight correction, which can better reflect the objectivity of the research [20]. Javier [21] and García [22] were the first to use DEA to calculate the service efficiency of the family health authority in the UK and the primary medical health center in Zaragoza, Spain, respectively, confirming the feasibility of DEA that this method can be used to evaluate input and output and effectively identify inefficient decision-making units in the primary medical health field. Since then, more scholars have applied DEA to study the efficiency of primary healthcare. For example, Nick has used DEA to measure the efficiency of primary healthcare institutions in remote areas of Greece [23] and has found that preventive medical services can greatly improve the service efficiency of primary medical institutions. Jon has used DEA to evaluate the work efficiency of primary doctors in the United States [24] and has found that more hospital days and fewer office visits could promote the efficiency of primary doctors in the United States.

With the elaboration of economy, primary healthcare also develops rapidly. Traditional DEA has many drawbacks, such as low efficiency measurement accuracy, which results in the inability to dynamically analyze efficiency changes and eliminate interference from external factors of the social environment. Therefore, in order to solve many problems, scholars further optimize the DEA model and put it into practice in the field of primary healthcare. For example, Yan has used the Malmquist index to measure the efficiency of primary healthcare after medical reform in China and has concluded that economically developed areas are more efficient [25]. Zhong has adopted the BCC-CCR model with variable returns to scale and introduced the Malmquist index to analyze the operation efficiency of primary medical health institutions in Hunan Province from dynamic and static perspectives, and has suggested that improving the technical ability and working enthusiasm of medical staff is the key to improve the operation efficiency of primary medical health institutions [26]. Mohammadpour has used DEA and Tobit regression to estimate the technical efficiency and determinants of primary healthcare in Hamadan, Iran, among which the number of midwives and the total fertility per population had obvious negative effect [27]. In order to improve the measurement accuracy, Oikonomou has revised the input-output indicators to measure the efficiency of primary medical health services in western and southern Greece, and has finally come to a similar but different conclusion with Nick; that is, most primary medical health institutions have strong potential, and paying attention to disease prevention and chronic disease management can significantly improve the efficiency of institutions [28]. To take the impact of external environmental factors on primary healthcare into consideration, Diedda has used a four-stage DEA model to analyze the operation efficiency of primary healthcare with the application of information and communication technology in the Basque Country of Spain, and has proposed that the government should measure external environmental factors in the corresponding region when making policies [29]. The relaxation of input and output is also critical, and the Slack-Based Measure model can be used to solve this problem [30]. However, most current studies use Slack-Based Measure to analyze the efficiency of public hospitals or medical health service systems, while few scholars use this method in the field of primary medical health.

Spatial characteristics of primary medical health have attracted many scholars. In terms of spatial accessibility and equity of primary medical health, Theil index [31], Gini coefficient [32], Coefficient of Variation [33], Spatial Auto-correlation [19,20], and other methods have been widely used. Over the past few years, more scholars have concentrated on the research field of spatial distribution of primary medical health institutions. The Gauss two-step mobile search method [34], Geo-Information System (GIS) [35,36], Seemingly Unrelated Regression (SUR) [37], and other quantitative methods have been used to comprehensively analyze the spatial distribution and service accessibility of primary medical health institutions. For example, Valorie has found that there is a significant north-south difference in the accessibility of doctors in primary health institutions in Canada, and the spatial distribution of doctors was consistent with the population distribution [38]. Tan has drawn the distribution map of primary medical health institutions in Hainan Province, has studied population measurement and spatial accessibility from different angles using Gaussian-based two-step floating catchment area method (2SFCA), and has finally emphasized that the identification function has a significant impact on the accessibility of primary medical services [39]. Spatial Markov probability transition matrix can also be applied to the medical health field [40] to reveal the impact of spatial factors on primary medical health care.

To sum up, this study can be extended and deepened from the following points. Firstly, in terms of research objects, most of the researchers have focused on hot issues such as public hospitals and medical health service systems rather than primary issues. Secondly, in efficiency measurement, most of the current studies have considered the relaxation of input-output from the perspective of public hospitals or health service systems, and few studies have analyzed it from the perspective of primary healthcare. Hence the research has further optimized the DEA model, has incorporated the relaxation of input-output into the research scope of primary healthcare, and has used super efficiency analysis to subsequently divide multiple decision-making units that are simultaneously effective. Thirdly, in terms of time and spatial research, most literature only discusses the development of primary healthcare from a single dimension of time or space, and lacks a comprehensive analysis of the combination of the two. The study used the traditional and spatial theory of Markov chain analysis to reveal the Spatial-temporal evolution of primary healthcare and apply limiting distribution to predict the future development of primary healthcare. Therefore, focusing on the supply efficiency of primary medical health services can expand relevant research in this field.

## 3. Methods

Based on the data of 20 central and western provinces from 2010 to 2019, this paper applies the super efficiency Slack-Based Measure model to calculate the supply efficiency of primary medical health services in the central and western regions, and then utilizes the Kernel density estimation to make a dynamic analysis of the time series from the overall level of the central and western regions. The dynamic evolution trend is further analyzed from the relative state of each province by the traditional Markov chain. Finally, the spatial Markov chain is constructed and the geographical spatial factors are incorporated into the research scope to discuss the spatial evolution of the supply efficiency of primary medical health services in the central and western regions.

The super efficiency Slack-Based Measure model analyzes the supply efficiency of primary medical health services in the central and western regions from the perspective of the relaxation of input-output, and further distinguishes multiple simultaneously effective decision-making units; Kernel density estimation discusses the supply efficiency of health services in the central and western regions as a whole. Markov chain analysis studies the independent point of view of each province, and the two have good complementarity from the whole to the part, and the spatial Markov chain analysis supplements the blank of space research. The three progress and complement each other, making the research more complete and comprehensive.

### 3.1. Super Efficiency Slack-Based Measure Model

Data Envelopment Analysis has first been proposed by Charnes A. and Cooper W.W. [41], and it is a non-parametric estimation method used to study relative efficiency. On the basis of DEA, Tone [42] introduces the concept of relaxation variables and proposes the Slack-Based Measure model to solve the relaxation problem of input-output variables. In order to distinguish the problem that multiple decision-making units are effective at the same time, Tone [43] has proposed the super efficiency Slack-Based Measure model in 2002. The super-efficient Slack-Based Measure model can not only analyze the supply efficiency of primary medical health services in the central and western regions from the perspective of the relaxation of input-output, but also further distinguish multiple effective decision-making units. Therefore, the super efficiency Slack-Based Measure model is introduced in this study to measure the supply efficiency of primary medical health services in 20 provinces in central and western China. The model is constructed as follows:(1)$$min\rho =\frac{1-\frac{1}{m}\sum_{i=1}^{m}\frac{s_{i}{}^{-}}{x_{ik}}}{1+\frac{1}{q}\sum_{r=1}^{q}\frac{s_{r}{}^{+}}{y_{rk}}}$$
$$s.t.\left\{\begin{array}{c} \overset{n}{\underset{j=1,j\neq k}{\sum }}x_{ij}\lambda_{j}+s_{i}{}^{-}\leq x_{ik}\\ \overset{n}{\underset{j=1,j\neq k}{\sum }}y_{ij}\lambda_{j}-s_{i}{}^{+}\geq y_{rk}\\ \lambda_{j},s_{i}{}^{+},s_{i}{}^{-}\geq 0\\ i=1,\ldots ,m;r=1,\ldots q;j=1,\ldots n\left(j\neq k\right) \end{array}\right.$$

In Formula (1): ρ is the supply efficiency of primary medical health services. ρ≥1 means that the supply efficiency of primary medical health services in central and western China is effective and can be further ranked. ρ<1 means that the supply efficiency of primary medical health services in central and Western China is invalid; n is the number of decision-making units (DMU), and in this paper it equals to 20, which means the number of selected provinces in central and western China; m, q are the quantity of input and output; $s_{i}{}^{+},s_{i}{}^{-}$ are the relaxation variable of input and output in the model; $\lambda_{j}$ is the weight vector of each decision-making unit; $x_{ik},\;y_{ik}$ are the input variable and output variable of each decision-making unit.

### 3.2. Kernel Density Estimation

Kernel density estimation, also known as non-parametric Kernel estimation [44], connects discrete variables through smooth curves and replaces histograms with continuous density curves to better describe the distribution of variables. Kernel density estimation can describe and analyze the supply efficiency of primary medical health services in the central and western regions from the overall level. The basic principle is that for a set of variable distributions, a continuous density curve can be used to describe the random variables. Assume that the density function of the random variable is  fx, and there are n independent observed values for the random distribution, which are $Y_{1},Y_{2}\ldots Y_{n}$, etc. Then the Kernel density estimation formula is:(2)$$f\left(x\right)=\frac{1}{nh}\overset{n}{\underset{j=1}{\sum }}k\left(\frac{Y_{i}-\overset{-}{Y}}{h}\right)$$

In the Formula (2), n is the number of the central and western provinces studied in this paper; h is the bandwidth; $Y_{i}$ represents the efficiency value of the central and western provinces of China; $k\left(\cdot \right)$ is the Kernel function. The Kernel function does not greatly affect the shape of the curve, but affects the degree of smoothness. The choice of bandwidth determines the shape of the curve. The larger the bandwidth, the smaller the variance of Kernel density estimation, and the optimal bandwidth generally equals to $h=cn^{-0.2}$ (c is constant) [45]. In this paper, a smoother Gaussian Kernel is adopted, and Kernel density estimation is used to explain the dynamic evolution of the supply efficiency of primary medical health services in the central and western regions from four perspectives [46]: the distribution position, shape, ductility, and polarization degree of the curve.

### 3.3. Markov Chain Theory

Kernel density estimation can only analyze the dynamic evolution from the whole level, but it cannot reveal the time process of a single province. Therefore, the traditional Markov chain has been introduced to explore the relative state transition of the central and western provinces, and the time evolution of the supply efficiency for primary medical health service has been meticulously grasped [47]. Markov is a random process of $\left\{X\left(t\right),t\in T\right\}$, the exponential set T of the random process corresponds to each period, and the finite state corresponds to the number of states for random variables. Then for all periods t and all possible states i,  j, they meet the condition of formula (3):(3)$$\begin{array}{c} P\left\{X\left(t\right)=j|X\left(t-1\right)=i,X\left(t-2\right)=i_{t-2},\ldots ,X\left(0\right)=i_{0}\right\}\\ =\left\{X\left(t\right)=j|X\left(t-1\right)=i\right\} \end{array}$$

Formula (3) shows the property of Markov chain; that is, the probability of random variable X being in j state at period t only depends on the state of X at period t−1 [48]. Let $X\left(t\right)=j$; that is, the state in period t is j, and all transition probabilities are $P_{ij}$, as shown in Formula (4):(4)$$P_{ij}=\frac{n_{ij}}{n_{j}}$$

In Formula (4), $P_{ij}$ represents the probability that the supply efficiency of primary medical health services in the central and western regions shifts from state i to state j; $n_{ij}$ represents the sum of the number of provinces that are in the state i in year t and transferred to the state j in the t+1 year; $n_{j}$ represents the sum of the number of provinces in the state j in all years. The supply efficiency of primary medical health services in central and western China is divided into N states, and the transfer probability matrix of N × N is constructed. Thus, the transition type and corresponding probability distribution of each state are analyzed by measuring the values ($P_{ij}$) of each element in the matrix. Then the study will grasp the transition trend and law of the supply efficiency of primary medical health services in the central and western regions and judge whether there is upward, downward, or unchanged transition, according to the type of service supply efficiency.

However, the traditional Markov chain cannot explain the influence of geographic spatial factors on the supply efficiency of primary medical health services in central and western China. Therefore, the concept of “spatial lag” is introduced on the basis of the traditional Markov chain, and the spatial Markov transition probability matrix is constructed to analyze the spatial evolution of the supply efficiency of primary medical health services in central and western China and to reveal the internal relationship among spatial factors. The specific method is to add a spatial lag term to the traditional Markov transition probability matrix (N × N) and decompose it into an N × N × N transition probability matrix, namely the spatial Markov transition probability matrix. $P_{ij}|N$ represents the probability that a province will transfer from state i to state j in the following year when the spatial lag is *N*.

Markov process will be in a stable state unaffected by time after a long time [49]. At this time, the stable state is also called the limiting distribution or stationary distribution of the Markov chain. Using this limiting distribution, it is possible to predict and evaluate the future development of the supply efficiency of primary medical health services in central and western China. The initial state is transferred n (n→∞) times according to the obtained traditional and spatial Markov transition probability matrix, and the balanced and stable state probability distribution is acquired. The calculation formula is as follows:(5)$$\underset{N\to \infty }{lim}\pi N=\underset{N\to \infty }{lim}\pi \left(N+1\right)=\pi$$

π represents the stationary distribution of the Markov chain process. If π satisfies $\sum_{i=1}^{n}\pi_{i}=1$ and $0\leq \pi_{i}\leq 1$, then π is the stationary distribution of the traditional Markov chain process.

In Formula (5), the calculation method of traditional Markov chain process can be extended to the spatial Markov chain to obtain the stationary distribution of the Markov process under different spatial lag states, so as to predict and analyze the spatial relationship of the supply efficiency of primary medical health services in central and western China. The technology road map of Markov chain theory is shown in Figure 1.

## 4. Data, Input-Output Selection

### 4.1. Data Resources

This study uses data from 2010 to 2019. Data are extracted from *China Healthcare Statistical Yearbook, China Health and Family Planning Statistical Yearbook,* and *China Health Statistical Yearbook* from 2011 to 2020. The central and western regions are divided according to the regional division standards of the 2020 *China Health Statistical Yearbook.* The central region includes 8 provinces of Shanxi, Jilin, Heilongjiang, Anhui, Jiangxi, Henan, Hubei, and Hunan; the western region covers 12 provinces, autonomous regions, and municipalities including Inner Mongolia, Chongqing, Guangxi, Sichuan, Guizhou, Yunnan, Tibet, Shaanxi, Gansu, Qinghai, Ningxia, and Xinjiang.

### 4.2. Input-Output Selection

The selection of input-output indicators has an important impact on the measurement results of super efficiency Slack-Based Measure analysis. Different scholars use different indicators to measure the efficiency of primary medical health services. For example, Liu has taken the number of community health service centers, the number of community health service technicians, and the number of beds as input indicators, and the amount of treatment, hospitalization, and bed utilization as output indicators [50]. Zhang takes total expenditure, the number of doctors, nurses, beds, and equipment cost over 10,000 yuan as input indicators, and the total income, number of discharged patients, the amount of treatment, and check-ups as output indicators [51]. Zhang takes the number of primary medical and health institutions and the number of beds and health personnel as input indicators, and the average amount of treatment and annual hospitalization rates as output indicators [52]. Wu takes the number of beds per thousand people, total value of medical equipment, number of doctors per thousand people, proportion of senior doctors, and proportion of doctors with a bachelor’s degree or above as input indicators, and the number of visits and bed utilization as output indicators [53]. Referring to the existing research and the maneuverability of the super-efficient Slack-Based Measure model [18,19,20,21], and consulting the opinions of experts of Health Commission of Sichuan Province Primary Health Department, Traditional Chinese Medicine Administration Department and research institutes such as National Health Commission Health Development Research Center, China Pharmaceutical University and Fudan University, this paper takes the number of primary medical and health institutions ($X_{1}^{*}$), the number of beds ($X_{2}^{*}$), and the number of healthcare personnel ($X_{3}^{*}$) as input indicators, the number of visits ($Y_{1}^{*}$), the number of admissions ($Y_{2}^{*}$), and the frequency of family healthcare service ($Y_{3}^{*}$) as output indicators. See Table 1 for details.

## 5. Results

### 5.1. Measurement of Supply Efficiency of Primary Medical Health Service in Central and Western China

Based on DEA-SOLVER Pro5 software (version 5.0., SAITECH Inc. Tokyo, Japan), this paper adopts the super-efficiency Slack-Based Measure model with non-radial and variable returns to scale to measure the supply efficiency of primary medical health services in 20 provinces in central and western China from 2010 to 2019. The results are shown in Table 2. In view of this foundation, the average value of each year is calculated, and shown as Figure 2, and the supply efficiency of primary medical health services in different regions is compared and analyzed.

The data results in Table 2 show that the efficiency values of Ningxia, Anhui, Sichuan, Henan, Hubei, Chongqing, Jiangxi, and Qinghai are all larger than 1 from 2010 to 2019, and they are in an effective state. Among them, the efficiency value of Ningxia ranks first every year with the highest efficiency value of 1.651 and the lowest of 1.425. The efficiency values of Shaanxi, Shanxi, Inner Mongolia, and Jilin are all less than 1 in each year from 2010 to 2019, and they are all in an invalid state. The lowest efficiency value occurs in Shaanxi in 2019, which is only 0.132. Heilongjiang, Gansu, and Xinjiang are all in effect for only one year from 2010 to 2019, and the years with effective states are 2010, 2010, and 2015 respectively. The efficiency value of other provinces fluctuates between valid and invalid states, and most of the invalid years are concentrated in 2015–2017, showing a “U” curve. It is found that there is a large gap in the supply efficiency of primary medical health services in central and western China, and the gap between the largest and the smallest efficiency reaches 1.519, far exceeding 1.

Combining the annual average of the supply efficiency of primary medical and health services in central and western China from 2010 to 2019, analysis is demonstrated in Figure 2. The results indicate that:

Firstly, as shown in Figure 2, the supply efficiency of primary medical health services in central and Western China manifests two stages from 2010 to 2019. The first is that from 2010 to 2014, the overall supply efficiency of primary medical health service in the central and western regions drops significantly, from the highest value of about 0.969 to about 0.850. In 2009, the Chinese government announces the “Opinions of the CPC Central Committee and The State Council on Deepening the Reform of the Medical Health System” to establish a basic medical health system covering urban and rural residents with the long-term goal of providing the public with safe, effective, convenient, and affordable medical health service. Since then, China has entered the stage of deepening the reform of the medical health system. In order to ensure the realization of basic medical health services for urban and rural residents, the Chinese government has issued a series of guiding policies, such as the improvement of the general practitioner system, the construction of rural doctors, the promotion and implementation of the hierarchical medical system, and the advancement of the contracted service of family doctors [54]. The implementation of policies has a lag. When an inherent mode system is broken, it will destroy the stability of the system to a certain extent and affect the normal operation of the industry [55]. China’s medical system reform in 2009 destroys the external stable environment. In the early stage of the alternating implementation of new and old policies, it would affect the normal operation of the primary medical health system to a certain extent. Therefore, from 2009 to 2014, there is a sudden decline in the supply efficiency of primary medical health services in central and western China. From 2015 to 2019, although the Chinese government continues to promulgate reform policies, it is more mature than the initial stage of reform, and the overall environment tends to be stable [56]. The fluctuation is 0.827 only in 2017. From 2014 to 2019, it shows a trend of gentle fluctuation as a whole, and the supply efficiency of primary medical health services in central and western China changes steadily. In a word, the sharp drop in the supply efficiency of medical health service from 2010 to 2014 has been caused by the destruction of the environmental system through the implementation of reforms, while from 2015 to 2019, due to the gradual recovery of the environmental system, the supply efficiency of service changes steadily, and the overall stability is stable.

Secondly, comparing the supply efficiency of primary medical health services in the central and western regions, it can be found that the trends of the two are generally similar, but there are some differences. The years of 2013 and 2014 are key turning points. From 2010 to 2013, the efficiency of the western region is significantly higher than that of the central region. From 2014 to 2017, the efficiency of the central region is higher than that of the western region, but the gap between the two does not change much and both saw a slight decline in 2017. After the year of 2018, the gap between the two rapidly narrowes. The reform of the medical health system in 2009 has a significant impact on the central and western regions, but the central region leads to show an upward turning point in 2014. It can be seen that the central region has more successful implementation of policies and a more mature medical health system than the western region. From 2014 to 2019, the supply efficiency of primary medical health services of both shows an overall stability and steadily changing trend. No matter from the perspective of the central and western regions as a whole or alone, the change trend of the two regions is similar, and the service supply efficiency shows an obvious downward trend in 2013 and 2017 with the overall consistency.

### 5.2. Series Analysis of the Supply Efficiency of Primary Medical Health Service in Central and Western China

By analyzing the measurement results of the supply efficiency of primary medical health services in the central and western regions, the research only obtains simple static conclusions, and cannot further describe the dynamic evolution and overall time series process of the central and western regions. Hence the study continues to explore the dynamic evolution process of the supply efficiency of primary medical health services in the central and western regions over time, apply the Kernel density estimation function of Gaussian normal distribution, and incorporate the key years 2013 and 2017 into the observation period. Thus, the estimated nuclear density curves of the central and western regions (Figure 3, Figure 4 and Figure 5) is drawn by intercepting 2011, 2013, 2015, 2017, and 2019 as the observation points, and the trends are hereby analyzed from the distribution location, shape, ductility, and polarization degree, respectively.

#### 5.2.1. Dynamic Evolution Process of Central and Western Regions

Firstly, the dynamic evolution of the overall supply efficiency of primary medical health services in the central and western regions is analyzed. Figure 3 plots the Kernel density estimation curve of the overall supply efficiency of primary medical health services in the central and western regions. It possesses the following characteristics during the observation period: Firstly, in the distribution position, the center position of the highest peak of the Kernel density curve slightly moves to the left, indicating that the supply efficiency of primary medical health services in the central and western regions has gradually evolved to a low level. Secondly, in terms of the shape of the curve, the peak value of the center of the curve gradually decreases during the observation period from high and sharp to wide and narrow, indicating that the degree of dispersion of the supply efficiency of primary medical health service in the central and western regions has increased and the absolute difference decreases gradually over time. Thirdly, from the ductility of the curve, it can be seen that during the observation period, the curve changes from an obvious right-trailing tail to a trailing disappearance, indicating that the difference in supply efficiency of primary medical health services in the central and western regions between the highest value (Ningxia) and the lowest region (Shaanxi, Inner Mongolia, Jilin, Shanxi) has decreased. Fourthly, from the analysis of the degree of polarization, the number of peaks and waves of the curve has always maintained a “double-peak” state during the observation period, indicating that there is a polarization phenomenon in the supply efficiency of primary medical health services in the central and western regions, but the degree of differentiation is gradually weakened during the observation period. The main peak-to-peak value of the double peaks shows a downward trend, and finally shows a weak polarization phenomenon. In general, the supply efficiency of primary medical health services in central and western China has been slowly declining, and the difference in the supply efficiency of primary medical health services among provinces has gradually decreased, forming a bipolar distribution of high and low clusters, but the degree of differentiation has weakened year by year.

#### 5.2.2. Dynamic Evolution Process of the Central Region

An independent analysis of the dynamic evolution of the central region is then carried out. Figure 4 plots the Kernel density estimation curve of the supply efficiency of primary medical health services in central China, which has the following characteristics during the observation period: Firstly, seen from the distribution position, the center position corresponding to the peak of the curve slightly shifts to the left, indicating that during the observation period, the supply efficiency of primary medical health services in the central region has slightly evolved to a low level. Secondly, from the analysis of the shape of the curve, the peak value of the curve decreases significantly with the peak width growing larger, indicating that the dispersion degree of the supply efficiency of primary medical health services in the central region has increased, and the absolute difference of service supply efficiency has decreased over time. Thirdly, from the view of ductility, the curve in the early observation period has an obvious left-trailing tail, and the trailing phenomenon disappears over time, indicating that the difference between the regions with the highest and the lowest supply efficiency of primary medical health services in the central region decreases. Fourthly, from the analysis of the degree of polarization, the number of peaks during the observation period shows a change of “single peaks-double peaks-triple peaks-insignificant double peaks”, which means that there is a certain gradient effect in the supply efficiency of primary medical health services in the central region. It has the characteristics of two-level or multi-level differentiation, but the main peak value gradually decreases during the observation period, indicating that the gradient effect is weakened. In general, the supply efficiency of primary medical health services in the central region has evolved to a low level, but compared with that of the overall central and western regions, the reduction is smaller and the difference between provinces is reduced. In addition, there is a multi-peak gradient effect, but the gradient effect is obviously weakened.

#### 5.2.3. Dynamic Evolution Process of Western Regions

Then the dynamic evolution of the western region is analyzed separately. Figure 5 plots the Kernel density estimation curve of the supply efficiency of primary medical health services in central China, which has the following characteristics during the observation period: Firstly, from the view of the distribution position, the center of the curve first moves significantly to the left as the years go by. Then it moves slightly to the right, but a trend of moving to the left as a whole is still shown, indicating that during the observation period, the supply efficiency of primary medical health services in the western region first evolves to a low level, then to a high level, but overall, it develops towards a low level. Secondly, judging from the shape of the curve, the peak value of the curve shows a downward trend, and the width of the main peak does not change significantly, which means that the degree of dispersion of the supply efficiency of primary medical health services in western region has increased during the observation period. Thirdly, from the view of ductility, during the observation period, the curve has an obvious right tail at the beginning, but it gradually disappears over time, indicating that the regional difference between the highest and lowest supply efficiency of primary medical health services in western region has decreased. Fourthly, from the analysis of the degree of polarization, the number of peaks shows a “double peak-single peak” change, which means that there is a gradient effect in the supply efficiency of primary medical health services in western region with the characteristics of two-level differentiation, but the double-peak pattern is broken over time. Therefore, the polarization phenomenon has disappeared, indicating that the degree of equalization of the supply efficiency of primary medical health services in western region has been improved. In general, in terms of efficiency, the supply efficiency of primary medical health services in the western region has significantly decreased, and in the central region it has decreased slightly with some years increased slightly. That is to say, the overall decrease in the supply efficiency of primary medical health services in the central and western regions is mainly caused by the decrease in the efficiency of the western region. From the perspective of the balance of development, there is always a significant gradient effect in the central region, but the map effect in the western region has been weakened and the differentiation phenomenon disappears in several years, indicating that compared with the central region, the development of the western region is relatively balanced, and the overall polar distribution in the central and western regions is mainly due to the uneven development of the central region.

### 5.3. Temporal Evolution Characteristics of the Supply Efficiency of Primary Medical Health Service in Central and Western Regions

According to the measurement analysis and time series analysis of the supply efficiency of primary medical health services in central and western regions, we can only analyze the dynamic evolution of central and western regions, and the evolution process of each province cannot be independently analyzed. Therefore, a traditional Markov chain has been constructed to analyze the temporal evolution of 20 provinces. Firstly, a Markov transition probability matrix is constructed, and the supply efficiency of primary medical health services in 20 provinces in central and western China from 2010 to 2019 is divided into four groups of state spaces according to the difference in the level of the natural division method, in order to maximize the difference of each group [57]. k=1,2,3,4 was used to represent the four state spaces, respectively. The larger the value of k, the greater the supply efficiency of primary medical health services in the region. The state space is divided into low state, medium-low state, medium-high state, and high state in turn. The result was shown in Table 3.

The elements on the main diagonal line (P_11_ → P_44_) in Table 3 represent the probability that the state type of the supply efficiency of primary medical health services in the province has not shifted, and the non-diagonal elements represent the probability that the state type of the supply efficiency of primary medical health services in the province will transfer. For example, P_11_ = 0.7872 indicates that the province has a probability of 0.7872 to maintain efficiency state type 1. P_12_ = 0.1702 indicates that the probability of the province to transferring from efficiency state type 1 to 2 is 0.1702. When the diagonal elements are greater than other values in the same line, it means that the province has the stability to keep its state unchanged. According to this, the temporal evolution of the supply efficiency of primary medical health services in central and western China can be obtained without considering the spatial relationship: Firstly, the supply efficiency of primary medical health services in central and western China has the stability of maintaining the original state. The elements of the main diagonal are P_11_ = 0.7872, P_22_ = 0.5172, P_33_ = 0.8353, and P_44_ = 0.7368, which are significantly larger than other non-diagonal elements, indicating that the supply efficiency of primary medical health services in the central and western regions in each state has a greater possibility of maintaining its own state remains unchanged, and there may be club convergence effects with different characteristics. Secondly, there is a possibility of cross-state transfer in the supply efficiency of primary medical health services in the central and western regions, but the probability of occurrence is small, and it is difficult to achieve. In all states, the probability of cross-state transition is smaller than that of adjacent state transition, such as P_12_ = 0.1702 < P_13_ = 0.0425, P_23_ = 0.1724 < P_24_ = 0, P_32_ = 0.0706 < P_31_ = 0.0353, it can be found that whether upward or downward, it is more difficult to do a cross-state transfer. The results show that the state transition of the supply efficiency of primary medical health services in the central and western regions is a gradual and relatively stable process, and it is difficult to achieve cross-state transition in a short period of time. Therefore, it is necessary to pay attention to top-level design and scientific development. Instead of increasing the investment of medical resources only, the sustainable investment and development of medical resources should be attached great importance to.

### 5.4. Spatial Evolution Characteristics of the Supply Efficiency of Primary Medical Health Service in Central and Western Regions

The efficiency of medical health services in a region is not only affected by itself, but also by the behavior of neighboring provinces [30]. In order to further analyze the spatial evolution differences of the supply efficiency of primary medical health services in 20 provinces in the central and western regions, this paper constructs a spatial Markov transition probability matrix to compare the element relationship between the two matrices. Thus, the study aims to explore the relationship between the probability of state transition of the supply efficiency of health service within provinces and neighboring provinces, considering the geographic spatial pattern.

The transition type diagram of the supply efficiency of primary medical health services in central and western China from 2010 to 2019 based on the traditional Markov transition probability matrix has been drawn. The starting point of observation is 2010 and the end point is 2019, and the state types were divided into upward, downward, and unchanged. Figure 6 shows that Gansu, Qinghai, Inner Mongolia, Heilongjiang, Henan, and Guizhou transfer downward, Sichuan, Anhui, and Hunan transfer upward, and the state types of other provinces remain unchanged. The provinces transferring downward are relatively concentrated, reflecting that the spatial evolution of the supply efficiency of primary medical health services in the central and western regions is related to the management level and technical level of the provinces.

According to Figure 7, it could be found that the supply efficiency of primary medical health services in central and western China are mainly on a downward trend, and there are always low-efficiency aggregation provinces such as Inner Mongolia, Heilongjiang, Jilin, Shaanxi, Shanxi, and high-efficiency aggregation provinces such as Hunan, Hubei, Henan, Anhui, Chongqing, and Sichuan. Ningxia has always been in high efficiency surrounded by low efficiency, while Guizhou has a spatial characteristic of low efficiency surrounded by high efficiency.

The traditional Markov chain only considers the time evolution of a specific province without considering the influence of adjacent regions on the province. Hereby a spatial Markov chain was introduced, and the concept of spatial lag was added to construct a spatial Markov transition probability matrix to study the spatial evolution characteristics of the supply efficiency of primary medical health services in central and western China with taking geographic spatial factors into account. The matrix result is shown as Table 4. By comparing the two transition probability matrices (Table 3 and Table 4), the spatial evolution characteristics of the supply efficiency of primary medical health services in the central and western regions could be obtained: Firstly, geographical spatial factors had a significant impact on the supply efficiency of primary medical health services in the central and western regions. Comparing the two matrices, it can be found that the transition probabilities of most elements change after the introduction of the spatial lag. Without considering the geographic spatial factors, P_12_ = 0.1702; after considering, P_12|1_ = 0.0909, P_12|2_ = 0.1333, P_12|3_ = 0.2353, P_12|4_ = 0.25. Therefore, it is crucial to consider geographic spatial factors in research. Secondly, the state stability of the supply efficiency of primary medical health services in the central and western regions would change with the type of spatial lag. The traditional Markov transition probability matrix shows that the stability of each state is P_11_ = 0.7872, P_22_ = 0.5172, P_33_ = 0.8353, P_44_ = 0.7368 when the geographic spatial factors are not considered. When the spatial lag is introduced and the state is 1, the stability probabilities are 0.9091, 0.6, 0, and 1, respectively; when the space lag state is 2, the stability probabilities are 0.8667, 0.5556, 0.8333, and 0.6667, respectively; when the space lag state is 3, the stability probabilities are 0.7059, 0.625, 0.8824, and 0.2; when the space lag state is 4, the stability probabilities are 0.5, 0.2857, 0.8, and 1, respectively. By comparison, it can be found that the stability of the supply efficiency of primary medical health services in the central and western regions in type 1 and 2 would deteriorate with the improvement of the spatial lag type; that is, the possibility of state transition would increase. Thirdly, high-state neighborhoods drive the improvement of the supply efficiency of primary medical health services in neighboring provinces. When geographic spatial factors are considered and adjacent to a province with a high state, its state transitioning upward has the most probability to increase. For example, when the spatial lag is 3, P_12|3_ = 0.2353 > P_12_ = 0.1702, P_13|3_ = 0.0589 > P_13_ = 0.0425. The results show that when the supply efficiency of primary medical health services in adjacent areas is high, the region has a certain positive spillover effect, and there will be a phenomenon of club convergence to a certain extent.

### 5.5. Trend Prediction of the Spatial-Temporal Evolution of the Supply Efficiency of Primary Medical Health Services in Central and Western Regions

The supply efficiency of primary medical health services in 20 provinces in central and western China has a certain stability, but the transition between different states still occurs, and the entire system has not reached a state of equilibrium. Markov’s limiting distribution means a final stable matrix after the supply efficiency of primary medical health services in each province in the central and western regions undergoing numerous state transitions on the basis of the initial state, according to the Markov transition probability matrix (Table 3 and Table 4). The obtained matrix distribution can be used to predict the long-term development of the supply efficiency of primary medical health services in the central and western regions.

Shown as Table 5, without considering the spatial lag, it can be found that there are more provinces in the type 1 and 2, and fewer provinces in the type 3 and 4. The total distribution of provinces in the sum of low and medium-low states (0.4993) is approximately equal to sum of the high and medium-high states (0.5002). The sum of the distributions of the state provinces forms an obvious double-peak pattern, and finally shows the phenomenon that the low state and the high state are clustered separately.

Considering the spatial lag, there are great differences in the limiting distribution of the supply efficiency of primary medical health services in the central and western regions. Comparing the four types of limiting distributions, it can be found that when adjacent to low-level provinces, the supply efficiency of primary medical health services converges to the low state, forming an obvious single-peak pattern, and the probability distribution of reaching the low state finally reaches 0.7251. When adjacent to the low-medium state, the final development is more balanced than the low state, but there is still 0.4237 in the low state, and the sum of the low and low-medium states reaches 0.6143, with a double-peak pattern, generally showing a phenomenon that the low value and the high value are clustered separately. When adjacent to the medium-high state, the probability distribution of reaching the medium-high state is 0.5008, which is significantly larger than the probability distribution of other state types. While the sum of the low and low-medium states still reaches 0.4454, that is, from the perspective of a single state type, it shows a single-peak convergence pattern. However, when the low and low-medium states are considered as a whole, it shows a double peak, with the low value and the high value clustered separately. When adjacent to the high state, the probability distribution of the final high state reaches 0.8243, much larger than any other type of state probability distribution, and finally shows a single-peak convergence pattern.

In general, without considering the spatial relationship, the supply efficiency of health services will have a downward trend in the next 10–20 years, and the problem of unbalanced development will still be prominent. The supply efficiency of primary medical health service in central and western China will show the distribution characteristics of high and low clusters. The spatial relationship will have a significant impact on the development in the next 10–20 years. For example, adjacent to the low and high states will develop into a limit distribution where low and high converge respectively. At this time, the supply efficiency of health service gradually tends to the state of adjacent provinces, and the degree of development balance will be greatly increased.

## 6. Conclusions and Discussion

Based on the data of 20 provinces in central and western China from 2010 to 2019, this paper studies the spatial-temporal evolution characteristics and trend prediction of the supply efficiency of primary medical health services. The super efficiency Slack-Based Measure model is used to measure and analyze the service supply efficiency, and the Kernel density estimation is used for time series analysis. Thus, the traditional and spatial Markov transition probability matrix is constructed to analyze the spatial-temporal evolution characteristics of the supply efficiency of primary medical health service in central and western regions and predict long-term development trends. The following conclusions are obtained.

Firstly, from the perspective of time, the supply efficiency of primary medical health services in the central and western regions has declined from 2010 to 2019, which is mainly due to the significant decline of efficiency in the western regions. Among them, the sharp decline in efficiency before 2014 is the result of the implementation of the reform of the medical health system, and after 2014, the efficiency fluctuates steadily. The degree of development balance of the whole central and western regions has increased and the gap between provinces has gradually decreased, but the problem of unbalanced development still emerges, especially in the central region, as the difference in efficiency would reach more than 10 times. There is always a certain gradient effect in the supply efficiency of primary medical health services in the central and western regions of China, showing a high-low respectively clustering effect. However, this aggregation characteristic is significantly weakening, which also reflects the improvement of the fairness of development. This conclusion is consistent with the research results of Yang [12] and Diedd [29] and confirms the Chinese government’s assessment of the state of primary medical health development in the central and western regions, that the development of primary medical health in central and western regions is insufficient and the problem of imbalance needs to be addressed.

Secondly, from the spatial perspective, the supply efficiency of primary medical health services in central and western China has the stability of keeping its own state unchanged. It is extremely difficult to achieve a large increase in efficiency in a short period of time, but the space effect affects this stability to a certain extent. This is similar to the conclusion of Arnaudo et al., [58] who has studied the spatial effect of health care in Argentina. When adjacent to a province with a high state, the probability of an upward shift in efficiency increases significantly, and it decreases when adjacent to a province with a low state. Therefore, this spatial relationship can better improve the supply efficiency of primary medical health services in the central and western regions. This conclusion is consistent with the findings of Chen [40] and Haschka [59] et al., namely, that policies need to consider spatial patterns of healthcare utilization to improve the allocation of medical resources. Among them, Sichuan, Anhui, and Hunan shift upward, while Gansu, Qinghai, Inner Mongolia, Heilongjiang, Henan, and Guizhou shift downward, showing certain cluster characteristics.

Thirdly, from the perspective of future development trend, under the long-term development of current state, the supply efficiency of primary medical health services in central and western China will decline in the next 10–20 years, eventually showing the distribution characteristics of high-low respectively clustering, and the problem of unbalanced development still emerges. When considering the impact of spatial relationship on future development, it is found that under the condition of low state or high state of spatial lag, the supply efficiency of health services will evolve and converge to a low level or a high level, finally showing two extreme convergence distributions, that is, extremely lower or higher efficiency, and the development balance will be greatly improved. This is similar to Liu’s research conclusion [60], but quite different from Chen’s [40] conclusion, which may be related to Chen’s drawing of different conclusions from a national perspective.

At present, scholars have explored the factors that lead to the spatial and temporal differences in the formation of primary medical health services from various perspectives, but most of them focus on the two aspects of population mobility and regional economic development. To further explore the factors of the spatial and temporal differences in the supply efficiency of primary medical health services in central and western China, this paper analyzes two aspects. The first is population mobility. Population movement has an important impact on the allocation of medical health resources in the region. The reduction of floating population during the COVID-19 epidemic has alleviated the problem of insufficient medical health resources [61]. Compared with the western region, the central region is more developed in industry, agriculture, and service industry. These developed industries attract the population of the western region to migrate [62]. With the progress of population migration, the central region needs to continuously adjust its medical health industry to support the normal medical needs of the floating population and ensure the basic medical health service of the people in the region. Primary medical health institutions are naturally valued as the first place to meet medical service. Therefore, compared with the western region, the service efficiency of the central region is higher, the efficiency recovery speed is also faster after the new medical reform, and it also has strong system stability. The second is regional economic development. The economic development level of the central region is much higher than that of the western region, and the higher level of economic development attracts more high-quality medical resources, such as doctors, nurses, professional care workers, etc. [63]. As a result, the scale and quality of primary medical health services in the central region are higher than those in the western region. In addition, due to differences in the economy and population, there is an inequitable distribution of medical health resources in various regions. Although the differences in the central and western regions have gradually decreased since the new medical reform in 2009, this pattern of economic size affecting medical health resources allocation has not changed [64].

Based on the above conclusions, the following policy recommendations are proposed. The main problem in the western region is the lack of development efficiency. Therefore, it is necessary to invest designedly and continuously in resources and accelerate the standardization construction of the primary medical health system. The problems in the central regions are low quality of development, so it is necessary to promote management level and the rational and equitable use of resources to improve the quality of primary medical health services. Each province needs to strengthen the construction of disciplines according to their own characteristic. For example, Tibet can take advantage of its location to actively promote the “Tibetan medicine” industry to make up for the lack of its own resources and improve service efficiency. As the most efficient province, Ningxia can build a cross-regional primary medical and health services construction platform and use information technology to share technical experience, relying on its own spatial spillover effect to drive the efficiency improvement of neighboring provinces. When formulating policies, it is necessary to pay attention to efficiency and maintaining the stability of its own state, instead of blindly and radically investing. The government also needs to attach great importance to top-level design and overall planning and respect the laws of scientific evolution to promote the high-quality development of primary medical and health services of the central and western regions.

There are still some limitations in the research. We only consider the evolution of time and space, and lack specific analysis of influencing factors and discussion on factors such as population flow and urban development. The research and analysis are carried out from a macroscopic and overall perspective, the characteristics of each province are not studied in detail, and there is a lack of detailed description of specific provinces. The follow-up will conduct in-depth research on the influencing factors and the characteristics of a single province.

## Figures and Tables

**Figure 1 ijerph-20-01664-f001:**
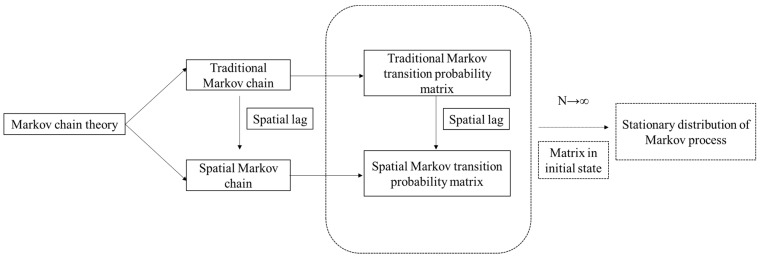
The Technology road map of Markov chain theory.

**Figure 2 ijerph-20-01664-f002:**
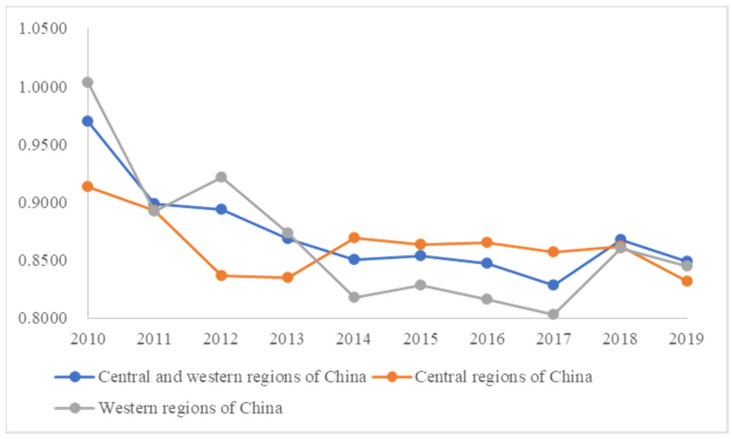
Annual average of primary medical and health service supply efficiency value in central and western China from 2010–2019.

**Figure 3 ijerph-20-01664-f003:**
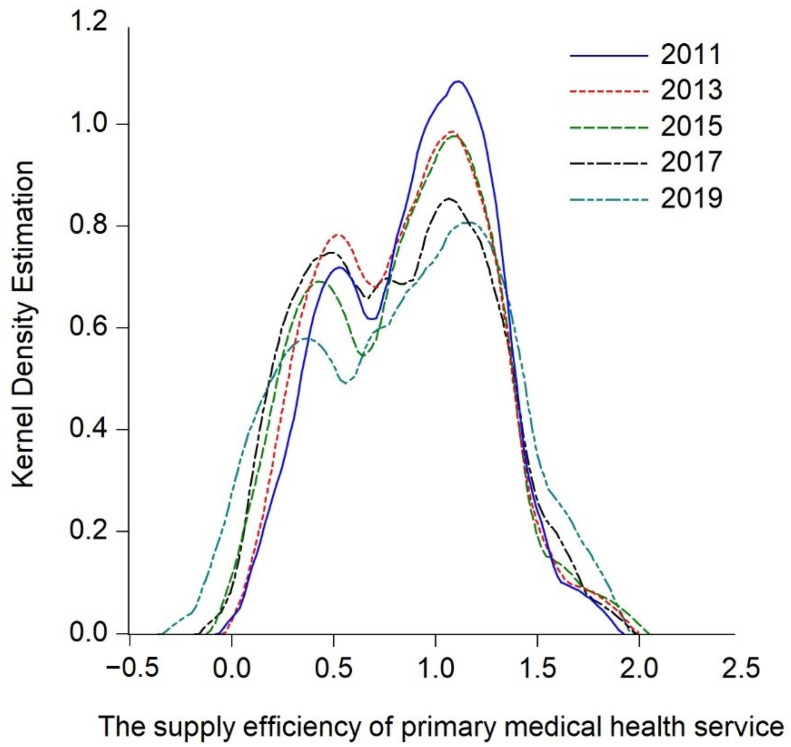
Kernel density estimation of the supply efficiency of primary medical health service in central and western China.

**Figure 4 ijerph-20-01664-f004:**
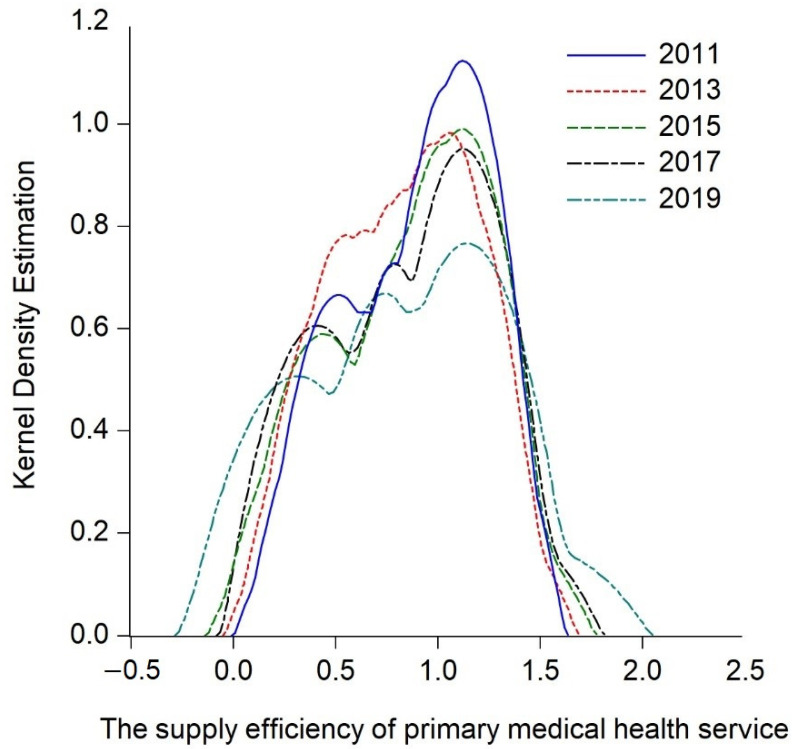
Kernel Density Estimation of the supply efficiency of primary healthcare services in central China.

**Figure 5 ijerph-20-01664-f005:**
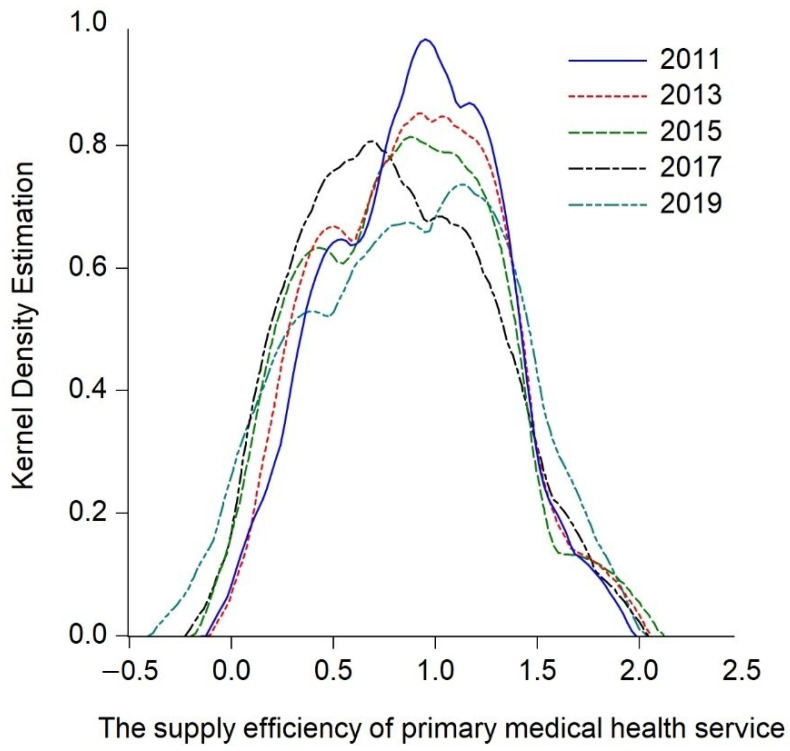
Kernel density estimation of the supply efficiency of primary medical health service in western China.

**Figure 6 ijerph-20-01664-f006:**
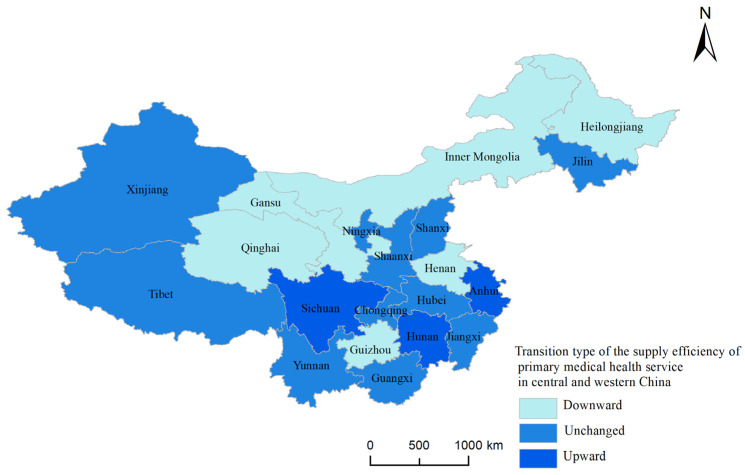
Transition type of the supply efficiency of primary medical health services in central and western China from 2010 to 2019.

**Figure 7 ijerph-20-01664-f007:**
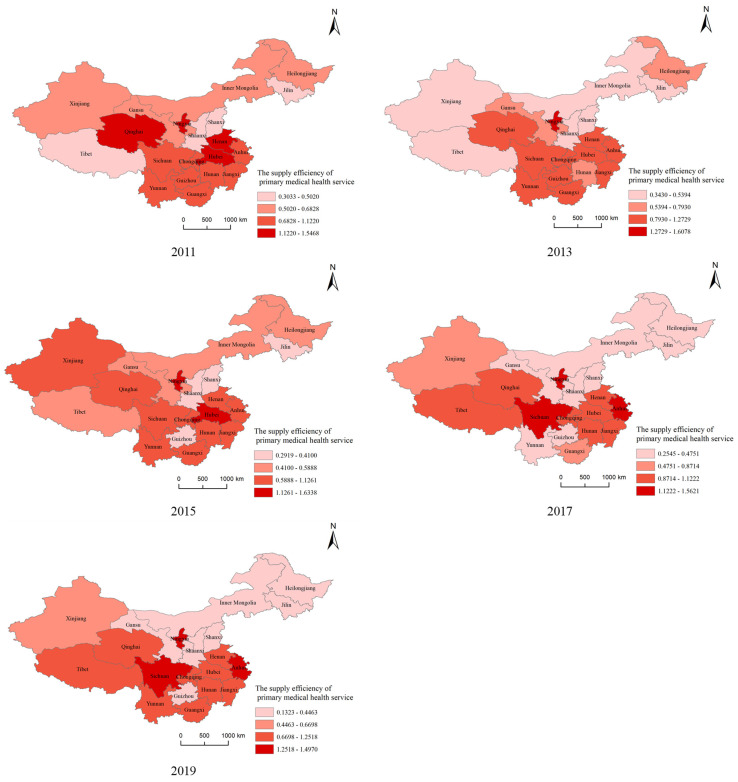
Distribution of supply efficiency of primary medical and health services in 20 provinces and cities in central and western China.

**Table 1 ijerph-20-01664-t001:** Input-output of the supply efficiency of primary medical and health services.

Indicator	Variable	Variable Description
Input indicators	The number of primary medical and health institutions (quantity)	Including community health service centers (stations), street health centers, township health centers, village clinics, outpatient departments, and clinics (infirmaries)
	The number of beds (quantity)	Annual fixed number of beds available (non-established beds)
	The number of healthcare personnel (quantity)	The number of employees working in primary health care facilities, including health technicians, village doctors and hygienists, other technicians, administrative staff, and commuters
Output indicators	The frequency of visits (10,000 person-times)	The total frequency of people visiting primary medical and health institutions for diagnosis and treatment
	The number of admissions (10,000 person)	Total inpatients in the reporting period
	The frequency of family healthcare service (10,000 person)	Number of visits by doctors to patients’ homes to provide medical, preventive and health-care services

**Table 2 ijerph-20-01664-t002:** Supply efficiency value of primary medical and health services in central and Western China from 2010 to 2019.

Province	2010	2011	2012	2013	2014	2015	2016	2017	2018	2019
Ningxia	1.425	1.547	1.538	1.608	1.651	1.634	1.586	1.562	1.492	1.497
Anhui	1.101	1.076	1.075	1.087	1.068	1.063	1.051	1.340	1.318	1.485
Sichuan	1.138	1.122	1.144	1.148	1.156	1.126	1.136	1.312	1.439	1.460
Henan	1.222	1.227	1.105	1.112	1.124	1.124	1.132	1.122	1.090	1.074
Hubei	1.158	1.209	1.299	1.273	1.312	1.316	1.310	1.081	1.091	1.087
Chongqing	1.047	1.076	1.086	1.052	1.088	1.082	1.090	1.098	1.081	1.082
Jiangxi	1.025	1.080	1.002	1.001	1.028	1.043	1.059	1.054	1.052	1.026
Qinghai	1.266	1.253	1.210	1.222	1.155	1.109	1.110	1.072	1.023	1.011
Hunan	0.756	1.011	0.797	0.793	1.023	1.041	1.040	1.020	1.062	1.038
Guangxi	1.057	1.014	1.037	1.092	1.069	1.049	1.035	0.871	1.001	1.025
Tibet	1.127	0.412	1.032	0.430	0.390	0.446	0.468	1.003	1.342	1.252
Yunnan	1.065	1.044	1.042	1.025	1.011	1.018	1.019	0.475	0.658	1.004
Guizhou	1.101	1.093	1.081	1.056	0.508	0.292	0.319	0.361	0.422	0.446
Xinjiang	0.779	0.683	0.624	0.539	0.640	1.001	0.815	0.756	0.733	0.670
Gansu	1.006	0.676	0.558	0.650	0.585	0.559	0.565	0.467	0.527	0.442
Heilongjiang	1.015	0.634	0.528	0.583	0.573	0.589	0.565	0.393	0.364	0.277
Jilin	0.516	0.405	0.420	0.369	0.381	0.323	0.351	0.426	0.466	0.366
Inner Mongolia	0.647	0.599	0.509	0.528	0.499	0.508	0.524	0.475	0.475	0.297
Shanxi	0.514	0.502	0.468	0.459	0.443	0.410	0.413	0.416	0.451	0.302
Shaanxi	0.428	0.303	0.315	0.343	0.307	0.336	0.348	0.255	0.270	0.132

**Table 3 ijerph-20-01664-t003:** Markov transition probability matrix of the supply efficiency of primary medical health services in central and western China from 2010 to 2019.

t/t + 1	1	2	3	4
1	0.7872	0.1702	0.0425	0
2	0.3103	0.5172	0.1724	0
3	0.0353	0.0706	0.8353	0.0588
4	0	0	0.2631	0.7368

**Table 4 ijerph-20-01664-t004:** Spatial Markov transition probability matrix of the supply efficiency of primary medical and health services in central and western China from 2010 to 2019.

State of Spatial Lag	t/t + 1	1	2	3	4
1	1	0.9091	0.0909	0	0
	2	0.4	0.6	0	0
	3	0	1	0	0
	4	0	0	0	1
2	1	0.8667	0.1333	0	0
	2	0.2222	0.5556	0.2222	0
	3	0.0417	0.0833	0.8333	0.0417
	4	0	0	0.3333	0.6667
3	1	0.7059	0.2353	0.0589	0
	2	0.375	0.625	0	0
	3	0	0.0294	0.8824	0.0882
	4	0	0	0.8	0.2
4	1	0.5	0.25	0.25	0
	2	0.2857	0.2857	0.4286	0
	3	0.08	0.08	0.8	0.04
	4	0	0	0	1

**Table 5 ijerph-20-01664-t005:** Trend prediction of the spatial-temporal evolution of the supply efficiency of primary medical health services in central and western China.

	State of Spatial Lag	1	2	3	4
Initial state		0.26	0.16	0.47	0.11
Limit distribution without considering spatial lag		0.3175	0.1718	0.4085	0.0917
Limit distribution considering spatial lag	1	0.7251	0.1649	0	0.11
	2	0.4237	0.1906	0.3388	0.0424
	3	0.2496	0.1958	0.5008	0.0552
	4	0.0381	0.0284	0.1248	0.8243

## Data Availability

The original contributions presented in the study are included in the article, further inquiries can be directed to the corresponding authors.

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
