# Peer review of "Temporal-Spatial Evolution and Trend Prediction of the Supply Efficiency of Primary Medical Health Service—An Empirical Study Based on Central and Western Regions of China"

_ijerph, 2023, doi:10.3390/ijerph20031664_

Round 1

Reviewer 1 Report

I read the manuscript with attention. It is an interested manuscript with high quality results. The statistical and methods were appropriate. The tables were clear and helpful.

 However, some comments are provided to improve the manuscript:

The introduction is written very long and can discourage the reader from reading it. In many parts of the introduction, additional explanations made the text heterogeneous. On the other hand, important explanations in the introduction have been ignored.

Show the necessity of proper management of COVID19 by referring to the following studies:

·        1.  Socioeconomic status and COVID-19-related cases and fatalities in the world: A cross-sectional ecological study

·       2.  The lost productivity cost of absenteeism due to COVID-19 in health care workers in Iran: a case study in the hospitals of Mashhad University of Medical Sciences

Explain about "the total abolition of drug addition" and say under what conditions it was implemented.

The literature review is well written and past studies are well referenced. But, writing an introduction need to the art of integrating the research field, literature review, the existing gap and stating the purpose of the study to fill this gap. You should artistically integrate the studies, especially the references 17 to 22, which are used in the literature review, in the introduction and discussion.

Author Response

Dear Reviewer 1

Return:

Thank you very much for your suggestions, which are of great help to the revision of our article. Therefore, we have made the following adjustments:

  1. In order to highlight the innovation of our research, we add literature discussion in related fields in Line96 - Line101 in the Introduction, and combine the research fields and literature review to highlight our research purpose and significance. At the same time, we have made detailed reference to the two articles you recommended. The discussion about medical staff, economic situation and COVID-19 in the articles plays an important role in the revision of our article. We refer the article The lost productivity cost of absenteeism due to COVID-19 in health care workers in Iran: a case study in the hospitals of Mashhad University of Medical Sciences you recommended based on the contents of the article.
  2. In order to enrich and integrate our research, we have specifically added references to the methods and results in the research fields in Line96 - Line113 in the Introduction. Meanwhile, we have also added specific conclusions of Literature Review in Line127 - Line176 to enrich the contents of literature review and make our conclusions more persuasive.
  3. In order to solve the problem of too long and disorderly introduction, we delete Line9 - Line103 in the original manuscript, and combine Line104 - Line111 in the original manuscript with the research fields and literature review, so as to make the article more clear.
  4. In order to explain "the abolition of drug addition", relevant descriptions are added to in Line65 - Line68. Specifically, "the abolition of drug addition" is to control drug costs and emphasize the public benefit attribute of hospitals. Chinese government pledged in 2017 to eliminate drug addition, which means that public hospitals at the county level and above will eliminate drug addition and sell some drugs at near-cost prices.

Best wishes

Reviewer 2 Report

Thank you for the opportunity to revise this manuscript. The topic is interesting and important, and addresses a core issue in any health system - the efficiency of the system and its potential drivers. 

The manuscript is well written, easy to follow, the design is appropriate, the analysis clearly presented and the results meaningful. Some comments and recommendations may improve the value of the work, as follows:

1. Lines 113-120: the authors aim to provide ground for their own definition of the supply efficiency of medical health service. I am wondering if the two authors mentioned on lines 114 (Samuelson) and 116 (Charnes) are the only perspectives to be discussed, and whether the authors' definition "this paper defines the supply efficiency of primary medical health service as the maximum of health service provided by primary medical health institutions with limited health resources." relates indeed with the ones proposed by the cited authors. I think a broader literature should be discussed regarding the relevance of the definition proposed in this paper. 

2. Lines 121-178: the authors propose a rich literature review in support to their choice of the super efficiency Slack-Based Measure to conduct the efficiency analysis involved here. Although there are good arguments regarding the limits of these alternative approaches, I would recommend more in depth discussions on the exact contexts and the results obtained using the Slack-Based Measure in the past. This part of the literature review should serve not only as an argument for the choice of the method, but also as a basis for further comparison of your own results with other results obtained based on the same method. 

3. In line with the above recommendation, the Discussions section is rich, and provides practical implications of your results. However, a comparison with results of other studies conducted with a similar methodology in other parts of the world may improve the quality of the paper. 

I will end up my review with congrats on a good and relevant piece of research. 

Author Response

Dear Reviewer 2

Return:

Thank you very much for your suggestions, which are of great help to the revision of literature review and conclusion and discussion of our article. We have made the following adjustments:

  1. As for the definition of the supply efficiency of medical health service, we believe that these two articles have good reference value in the preliminary review. But as you suggested, it has some limitation of readers' thinking to list only two authors' citations. Therefore, Sengupta's research and documents published by WHO, which are closely related to the definition of this study, are added in Line119 - Line123, serving as a helpful supplement to the relevance of our definition.
  2. As you said, the addition of research background and conclusions in the Literature Review helps to provide the basis for comparison of our conclusions, which is also an indispensable part of the discussion. Therefore, we enrich the elaboration of the background and conclusions in the Literature Review, specifically in Line143 - Line147 and Line154 - Line158. These conclusions will be of great help to our discussion.
  3. Thank you very much for your suggestions on Conclusion and discussion. We have added the comparison of research results in different regions with similar methods to improve the quality of our paper, which is specifically in Line737 - Line738, Line746 - Line748, Line751 - Line754 and Line765 - Line768. On the whole, our conclusion has similar conclusions with those researches, which also provides a strong reference for our conclusion.

Best wishes

Reviewer 3 Report

Dear Sir, 

After review of the paper titled "Temporal-Spatial Evolution and Trend Prediction of the Supply Efficiency of Primary Medical Health Service—An empirical study based on central and western regions of China" which submitted to the International Journal of

Environmental Research and Public Health, I think that this paper is suitable for publication in this Journal.

This paper need minor revisions to will improved and I think that the authors can consider it. So, the authors can:

- Add several recent references from 2021 to 2023.

- Revise some grammatical mistakes

- Explain some Chinese sentences in English language.

After that these revisions are approved, I think that the editor can consider this paper for publication>

All the best

Author Response

Dear Reviewer 3

Return:

Thank you very much for your suggestions. Your suggestions are greatly helpful for the revision of our article, especially for the grammar errors, which make our article to be more accurate. We propose the following modifications:

  1. We have enriched and updated part of recent articles from 2021 to 2023 to make our research closer to the frontier field. Among the references, 10, 11, 12, 13, 16, 17, 58, 59 and 60 are the latest articles from 2021 to 2023.
  2. We have corrected some grammar errors, specifically in Line9 - Line29, Line115 - Line117 and Line717 - Line813. Thank you for your questions and correcting these grammar errors make our article more accurate.
  3. In view of some Chinese explanations, we have made some modifications and make more accurate expressions in English, specifically in Line65 - Line68, Line115 - Line126 and other parts.

Best wishes

Round 2

Reviewer 2 Report

The authors addressed all my comments and requests.